# Change-point Detection for Sparse and Dense Functional Data in General Dimensions

**Carlos Misael Madrid Padilla**
Department of Mathematics
University of Notre Dame
cmadridp@nd.edu

**Daren Wang**
Department of Statistics
University of Notre Dame
dwang24@nd.edu

**Zifeng Zhao**
Mendoza College of Business
University of Notre Dame
zzhao2@nd.edu

**Yi Yu**
Department of Statistics
University of Warwick
yi.yu.2@warwick.ac.uk

## Abstract

We study the problem of change-point detection and localisation for functional data sequentially observed on a general $d$-dimensional space, where we allow the functional curves to be either sparsely or densely sampled. Data of this form naturally arise in a wide range of applications such as biology, neuroscience, climatology and finance. To achieve such a task, we propose a kernel-based algorithm namely functional seeded binary segmentation (FSBS). FSBS is computationally efficient, can handle discretely observed functional data, and is theoretically sound for heavy-tailed and temporally-dependent observations. Moreover, FSBS works for a general $d$-dimensional domain, which is the first in the literature of change-point estimation for functional data. We show the consistency of FSBS for multiple change-point estimation and further provide a sharp localisation error rate, which reveals an interesting phase transition phenomenon depending on the number of functional curves observed and the sampling frequency for each curve. Extensive numerical experiments illustrate the effectiveness of FSBS and its advantage over existing methods in the literature under various settings. A real data application is further conducted, where FSBS localises change-points of sea surface temperature patterns in the south Pacific attributed to El Niño. The code to replicate all of our experiments can be found at `https://github.com/cmadridp/FSBS`.

## 1 Introduction

Recent technological advancement has boosted the emergence of functional data in various application areas, including neuroscience [e.g. 11, 23], finance [e.g. 13], transportation [e.g. 10], climatology [e.g. 7, 14] and others. We refer the readers to [29] - a comprehensive review, for recent development of statistical research in functional data analysis.

In this paper, we study the problem of change-point detection and localisation for functional data, where the data are observed sequentially as a time series and the mean functions are piecewise stationary, with abrupt changes occurring at unknown time points. To be specific, denote $\mathcal{D}$ as a general $d$-dimensional space that is homeomorphic to $[0, 1]^d$, where $d \in \mathbb{N}^+$ is considered as arbitrary but fixed. We assume that the observations $\{(x_{t,i}, y_{t,i})\}_{t=1,i=1}^{T,n} \subseteq \mathcal{D} \times \mathbb{R}$ are generated based on

$$y_{t,i} = f_t^*(x_{t,i}) + \xi_t(x_{t,i}) + \delta_{t,i}, \text{ for } t = 1, \ldots, T \text{ and } i = 1, \ldots, n. \tag{1}$$

36th Conference on Neural Information Processing Systems (NeurIPS 2022).

In this model, $\{x_{t,i}\}_{t=1,i=1}^{T,n} \subseteq \mathcal{D}$ denotes the discrete grids where the (noisy) functional data $\{y_{t,i}\}_{t=1,i=1}^{T,n} \subseteq \mathbb{R}$ are observed, $\{f_t^* : \mathcal{D} \to \mathbb{R}\}_{t=1}^T$ denotes the deterministic mean functions, $\{\xi_t : \mathcal{D} \to \mathbb{R}\}_{t=1}^T$ denotes the functional noise and $\{\delta_{t,i}\}_{t=1,i=1}^{T,n} \subseteq \mathbb{R}$ denotes the measurement error. We refer to Assumption 1 below for detailed technical conditions on the model.

To model the unstationarity of sequentially observed functional data which commonly exists in real world applications, we assume that there exist $K \in \mathbb{N}$ change-points, namely $0 = \eta_0 < \eta_1 < \cdots < \eta_K < \eta_{K+1} = T$, satisfying that $f_t^* \neq f_{t+1}^*$, if and only if $t \in \{\eta_k\}_{k=1}^K$. Our primary interest is to accurately estimate $\{\eta_k\}_{k=1}^K$.

Due to the importance of modelling unstationary functional data in various scientific fields, this problem has received extensive attention in the statistical change-point literature, see e.g. [3], [6], [17], [31], [4] and [12]. Despite the popularity, we identify a few limitations in the existing works. Firstly, both the methodological validity and theoretical guarantees of all these papers require fully observed functional data without measurement error, which may not be realistic in practice. Secondly, most existing works focus on the single change-point setting and to our best knowledge, there is no consistency result of multiple change-point estimation for functional data. Lastly but most importantly, existing algorithms only consider functional data with support on $[0, 1]$ and thus are not applicable to functional data with multi-dimensional domain, a type of data frequently encountered in neuroscience and climatology.

In view of the aforementioned three limitations, in this paper, we make several theoretical and methodological contributions, summarized below.

• In terms of methodology, our proposed kernel-based change-point detection algorithm, functional seeded binary segmentation (FSBS), is computationally efficient, can handle discretely observed functional data contaminated with measurement error, and allows for temporally-dependent and heavy-tailed data. FSBS, in particular, works for a general $d$-dimensional domain with arbitrary but fixed $d \in \mathbb{N}^+$. This level of generality is the first time seen in the literature.

• In terms of theory, we show that under standard regularity conditions, FSBS is consistent in detecting and localising multiple change-points. We also provide a sharp localisation error rate, which reveals an interesting phase transition phenomenon depending on the number of functional curves observed $T$ and the sampling frequency for each curve $n$. To the best of our knowledge, the theoretical results we provide in this paper are the sharpest in the existing literature.

• A striking case we handle in this paper is that each curve is only sampled at one point, i.e. $n = 1$. To the best of our knowledge, all the existing functional data change-point analysis papers assume full curves are observed. We not only allow for discrete observation, but carefully study this most extreme sparse case $n = 1$ and provide consistent localisation of the change-points.

• We conduct extensive numerical experiments on simulated and real data. The result further supports our theoretical findings, showcases the advantages of FSBS over existing methods and illustrates the practicality of FSBS.

• A byproduct of our theoretical analysis is new theoretical results on kernel estimation for functional data under temporal dependence and heavy-tailedness. This set of new results *per se* are novel, enlarging the toolboxes of functional data analysis.

**Notation and definition.** For any function $f : [0,1]^d \to \mathbb{R}$ and for $1 \le p < \infty$, define $\|f\|_p = (\int_{[0,1]^d} |f(x)|^p \, \mathrm{d}x)^{1/p}$ and for $p = \infty$, define $\|f\|_\infty = \sup_{x \in [0,1]^d} |f(x)|$. Define $\mathcal{L}_p = \{f : [0,1]^d \to \mathbb{R}, \|f\|_p < \infty\}$. For any vector $s = (s_1, \ldots, s_d)^\top \in \mathbb{N}^d$, define $|s| = \sum_{i=1}^d s_i$, $s! = s_1! \cdots s_d!$ and the associated partial differential operator $D^s = \frac{\partial^{|s|}}{\partial x_1^{s_1} \cdots \partial x_d^{s_d}}$. For $\alpha > 0$, denote $\lfloor \alpha \rfloor$ to be the largest integer smaller than $\alpha$. For any function $f : [0,1]^d \to \mathbb{R}$ that is $\lfloor \alpha \rfloor$-times continuously differentiable at point $x_0$, denote by $f_{x_0}^\alpha$ its Taylor polynomial of degree $\lfloor \alpha \rfloor$ at $x_0$, which is defined as $f_{x_0}^\alpha(x) = \sum_{|s| \le \lfloor \alpha \rfloor} \frac{(x-x_0)^s}{s!} D^s f(x_0)$. For a constant $L > 0$, let $\mathcal{H}^\alpha(L)$ be the set of functions $f : [0,1]^d \to \mathbb{R}$ such that $f$ is $\lfloor \alpha \rfloor$-times differentiable for all $x \in [0,1]^d$ and satisfy $|f(x) - f_{x_0}^\alpha(x)| \le L|x - x_0|^\alpha$, for all $x, x_0 \in [0,1]^d$. Here $|x - x_0|$ is the Euclidean distance between $x, x_0 \in \mathbb{R}^d$. In non-parametric statistical literature, $\mathcal{H}^\alpha(L)$ are often referred to as the class

of Hölder smooth functions. We refer the interested readers to [25] for more detailed discussion on Hölder smooth functions.

For two positive sequences $\{a_n\}_{n\in\mathbb{N}^+}$ and $\{b_n\}_{n\in\mathbb{N}^+}$, we write $a_n = O(b_n)$ or $a_n \lesssim b_n$ if $a_n \leq Cb_n$ with some constant $C > 0$ that does not depend on $n$, and $a_n = \Theta(b_n)$ or $a_n \asymp b_n$ if $a_n = O(b_n)$ and $b_n = O(a_n)$.

## 2 Functional seeded binary segmentation

### 2.1 Problem formulation

Detailed model assumptions imposed on model (1) are collected in Assumption 1. For notational simplicity, without loss of generality, we set the general $d$-dimensional domain $\mathcal{D}$ to be $[0,1]^d$, as the results apply to any $\mathcal{D}$ that is homeomorphic to $[0,1]^d$.

**Assumption 1.** *The data $\{(x_{t,i}, y_{t,i})\}_{t=1,i=1}^{T,n} \subseteq [0,1]^d \times \mathbb{R}$ are generated based on model* (1).

**a.** *(Discrete grids) The grids $\{x_{t,i}\}_{t=1,i=1}^{T,n} \subseteq [0,1]^d$ are independently sampled from a common density function $u : [0,1]^d \to \mathbb{R}$. In addition, there exist constants $r > 0$ and $L > 0$ such that $u \in \mathcal{H}^r(L)$ and that $\inf_{x\in[0,1]^d} u(x) \geq \tilde{c}$ with an absolute constant $\tilde{c} > 0$.*

**b.** *(Mean functions) For $r > 0$ and $L > 0$, we have $f_t^* \in \mathcal{H}^r(L)$. The minimal spacing between two consecutive change-points $\Delta = \min_{k=1}^{K+1}(\eta_k - \eta_{k-1})$ satisfies that $\Delta = \Theta(T)$.*

**c.** *(Functional noise) Let $\{\varepsilon_i, \varepsilon_0'\}_{i\in\mathbb{Z}}$ be i.i.d. random elements taking values in a measurable space $S_\xi$ and $g$ be a measurable function $g : S_\xi^\infty \to \mathcal{L}_2$. The functional noise $\{\xi_t\}_{t=1}^T \subseteq \mathcal{L}_2$ takes the form*

$$\xi_t = g(\mathcal{G}_t), \quad \text{with } \mathcal{G}_t = (\ldots, \varepsilon_{-1}, \varepsilon_0, \varepsilon_1, \ldots, \varepsilon_{t-1}, \varepsilon_t).$$

*There exists an absolute constant $q \geq 3$, such that $\mathbb{E}(\|\xi_t\|_\infty^q) < C_{\xi,1}$ for some absolute constant $C_{\xi,1}$. Define a coupled process*

$$\xi_t^* = g(\mathcal{G}_t^*), \quad \text{with } \mathcal{G}_t^* = (\ldots, \varepsilon_{-1}, \varepsilon_0', \varepsilon_1, \ldots, \varepsilon_{t-1}, \varepsilon_t).$$

*We have $\sum_{t=1}^\infty t^{1/2-1/q}\{\mathbb{E}\|\xi_t - \xi_t^*\|_\infty^q\}^{1/q} < C_{\xi,2}$ for some absolute constant $C_{\xi,2} > 0$.*

**d.** *(Measurement error) Let $\{\epsilon_i, \epsilon_0'\}_{i\in\mathbb{Z}}$ be i.i.d. random elements taking values in a measurable space $S_\delta$ and $\tilde{g}_n$ be a measurable function $\tilde{g}_n : S_\delta^\infty \to \mathbb{R}^n$. The measurement error $\{\delta_t\}_{t=1}^T \subseteq \mathbb{R}^n$ takes the form*

$$\delta_t = \tilde{g}_n(\mathcal{F}_t), \quad \text{with } \mathcal{F}_t = (\ldots, \epsilon_{-1}, \epsilon_0, \epsilon_1, \ldots, \epsilon_{t-1}, \epsilon_t).$$

*There exists an absolute constant $q \geq 3$, such that $\max_{i=1}^n \mathbb{E}(|\delta_{t,i}|^q) < C_{\delta,1}$ for some absolute constant $C_{\delta,1}$. Define a coupled process*

$$\delta_t^* = \tilde{g}_n(\mathcal{F}_t^*), \quad \text{with } \mathcal{F}_t^* = (\ldots, \epsilon_{-1}, \epsilon_0', \epsilon_1, \ldots, \epsilon_{t-1}, \epsilon_t).$$

*We have $\max_{i=1}^n \sum_{t=1}^\infty t^{1/2-1/q}\{\mathbb{E}|\delta_{t,i} - \delta_{t,i}^*|^q\}^{1/q} < C_{\delta,2}$ for some absolute constant $C_{\delta,2} > 0$.*

Assumption 1**a** allows the functional data to be observed on discrete grids and moreover, we allow for different grids at different time points. The sampling distribution $\mu$ is required to be lower bounded on the support $[0,1]^d$, which is a standard assumption widely used in the nonparametric literature [e.g. 26]. Here, different functional curves are assumed to have the same number of grid points $n$. We remark that this is made for presentation simplicity only. It can indeed be further relaxed and the main results below will then depend on both the minimum and maximum numbers of grid points.

Note that Assumption 1**a** does not impose any restriction between the sampling frequency $n$ and the number of functional curves $T$, and indeed our method can handle both the dense case where $n \gg T$ and the sparse case where $n$ can be upper bounded by a constant. Besides the random sampling scheme studied here, another commonly studied scenario is the fixed design, where it usually assumes that the sampling locations $\{x_i\}_{i=1}^n$ are common to all functional curves across time. We remark that while we focus on the random design here, our proposed algorithm can be directly applied to the fixed design case without any modification. Furthermore, its theoretical justification under the fixed design case can be established similarly with minor modifications, which is omitted.

The observed functional data have mean functions $\{f_t^*\}_{t=1}^T$, which are assumed to be Hölder continuous in Assumption 1**b**. Note that the Hölder parameters in Assumption 1**a** and **b** are both denoted by $r$. We remark that different smoothness are allowed and we use the same $r$ here for notational simplicity. This sequence of mean functions is our primary interest and is assumed to possess a piecewise constant pattern, with the minimal spacing $\Delta$ being of the same order as $T$. This assumption essentially requires that the number of change-points is upper bounded. It can also be further relaxed and we will have more elaborated discussions on this matter in Section 5.

Our model allows for two sources of noise - functional noise and measurement error, which are detailed in Assumption 1**c** and **d**, respectively. Both the functional noise and the measurement error are allowed to possess temporal dependence and heavy-tailedness. For temporal dependence, we adopt the physical dependence framework by [30], which covers a wide range of time series models, such as ARMA and vector AR models. It further covers popular functional time series models such as functional AR and MA models [17]. We also remark that Assumption 1**c** and **d** impose a short range dependence, which is characterized by the absolute upper bounds $C_{\xi,2}$ and $C_{\delta,2}$. Further relaxation is possible by allowing the upper bounds $C_{\xi,2}$ and $C_{\delta,2}$ to vary with the sample size $T$.

The heavy-tail behavior is encoded in the parameter $q$. In Assumption 1**c** and **d**, we adopt the same quantity $q$ for presentational simplicity and remark that different heavy-tailedness levels are allowed. An extreme example is that when $q = \infty$, the noise is essentially sub-Gaussian. Importantly, Assumption 1**d** does not impose any restriction on the cross-sectional dependence among measurement errors observed on the same time $t$, which can be even perfectly correlated.

## 2.2 Kernel-based change-point detection

To estimate the change-point $\{\eta_k\}_{k=1}^K$ in the mean functions $\{f_t^*\}_{t=1}^T$, we propose a kernel-based cumulative sum (CUSUM) statistic, which is simple, intuitive and computationally efficient. The key idea is to recover the unobserved $\{f_t^*\}_{t=1}^T$ from the observations $\{(x_{t,i}, y_{t,i})\}_{t=1,i=1}^{T,n}$ based on kernel estimation.

Given a kernel function $K(\cdot) : \mathbb{R}^d \to \mathbb{R}^+$ and a bandwidth parameter $h > 0$, we define $K_h(x) = h^{-d} K(x/h)$ for $x \in \mathbb{R}^d$. Given the random grids $\{x_{t,i}\}_{t=1,i=1}^{T,n}$ and a bandwidth parameter $\bar{h}$, we define the density estimator of the sampling distribution $u(x)$ as

$$\hat{p}(x) = \hat{p}_{\bar{h}}(x) = \frac{1}{nT} \sum_{t=1}^T \sum_{i=1}^n K_{\bar{h}}(x - x_{t,i}), \quad x \in [0,1]^d.$$

Given $\hat{p}(x)$ and a bandwidth parameter $h > 0$, for any time $t = 1, 2, \cdots, T$, we define the kernel-based estimation for $f_t^*(x)$ as

$$F_{t,h}(x) = \frac{\sum_{i=1}^n y_{t,i} K_h(x - x_{t,i})}{n\hat{p}(x)}, \quad x \in [0,1]^d. \tag{2}$$

Based on the kernel estimation $F_{t,h}(x)$, for any integer pair $0 \le s < e \le T$, we define the CUSUM statistic as

$$\widetilde{F}_{t,h}^{(s,e)}(x) = \sqrt{\frac{e-t}{(e-s)(t-s)}} \sum_{l=s+1}^t F_{l,h}(x) - \sqrt{\frac{t-s}{(e-s)(e-t)}} \sum_{l=t+1}^e F_{l,h}(x), \quad x \in [0,1]^d. \tag{3}$$

The CUSUM statistic defined in (3) is the cornerstone of our algorithm and is based on two kernel estimators $\hat{p}(\cdot)$ and $F_{t,h}(\cdot)$. At a high level, the CUSUM statistic $\widetilde{F}_{t,h}^{(s,e)}(\cdot)$ estimates the difference in mean between the functional data in the time intervals $(s, t]$ and $(t, e]$. In the functional data analysis literature, other popular approaches for mean function estimation are reproducing kernel Hilbert space based methods and local polynomial regression. However, to our best knowledge, existing works based on the two approaches typically require that the functional data are temporally independent and it is not obvious how to extend their theoretical guarantees to the temporal dependence case. We therefore choose the kernel estimation method owing to its flexibility in terms of both methodology and theory and we derive new theoretical results on kernel estimation for functional data under temporal dependence and heavy-tailedness. We would like to point out that in the existing literature,

kernel-based change-point estimation methods are used in detecting change-points in nonparametric models [e.g. 20, 1, 21, 22].

For multiple change-point estimation, a key ingredient is to isolate each single change-point with well-designed intervals in $[0, T]$. To achieve this, we combine the CUSUM statistic in (3) with a modified version of the seeded binary segmentation (SBS) proposed in [19]. SBS is based on a collection of deterministic intervals defined in Definition 1.

**Definition 1** (Seeded intervals). *Let $\mathcal{K} = \lceil C_\mathcal{K} \log \log(T) \rceil$, with some sufficiently large absolute constant $C_\mathcal{K} > 0$. For $k \in \{1, \ldots, \mathcal{K}\}$, let $\mathcal{J}_k$ be the collection of $2^k - 1$ intervals of length $l_k = T2^{-k+1}$ that are evenly shifted by $l_k/2 = T2^{-k}$, i.e.*

$$\mathcal{J}_k = \{(\lfloor (i-1)T2^{-k} \rfloor, \lceil (i-1)T2^{-k} + T2^{-k+1} \rceil], \quad i = 1, \ldots, 2^k - 1\}.$$

*The overall collection of seeded intervals is denoted as $\mathcal{J} = \cup_{k=1}^{\mathcal{K}} \mathcal{J}_k$.*

The essential idea of the seeded intervals defined in Definition 1 is to provide a multi-scale system of searching regions for multiple change-points. SBS is computationally efficient with a computational cost of the order $O(T \log(T))$ [19].

Based on the CUSUM statistic and seeded intervals, Algorithm 1 summarises the proposed functional seeded binary segmentation algorithm (FSBS) for multiple change-point estimation in sequentially observed functional data. There are three main tuning parameters involved in Algorithm 1, the kernel bandwidth $\bar{h}$ in the estimation of the sampling distribution, the kernel bandwidth $h$ in the estimation of the mean function and the threshold parameter $\tau$ for declaring change-points. Their theoretical and numerical guidance will be presented in Sections 3.1 and 4, respectively.

---

**Algorithm 1** Functional Seeded Binary Segmentation. FSBS $((s, e], \bar{h}, h, \tau)$

---

**INPUT:** Data $\{x_{t,i}, y_{t,i}\}_{t=1, i=1}^{T, n}$, seeded intervals $\mathcal{J}$, tuning parameters $\bar{h}, h, \tau > 0$.

  **Initialization**: If $(s, e] = (0, n]$, set $\mathbf{S} \leftarrow \varnothing$ and set $\rho \leftarrow \log(T) n^{-1} h^{-d}$. Furthermore, sample $\lceil \log(T) \rceil$ points from $\{x_{t,i}\}_{t=1, i=1}^{T, n}$ uniformly at random without replacement and denote them as $\{u_m\}_{m=1}^{\lceil \log(T) \rceil}$. Estimate the sampling distribution evaluated at $\{\hat{p}_{\bar{h}}(u_m)\}_{m=1}^{\lceil \log(T) \rceil}$.

  **for** $\mathcal{I} = (\alpha, \beta] \in \mathcal{J}$ and $m \in \{1, \ldots, \lceil \log(T) \rceil\}$ **do**

    **if** $\mathcal{I} = (\alpha, \beta] \subseteq (s, e]$ and $\beta - \alpha > 2\rho$ **then**

      $A_m^{\mathcal{I}} \leftarrow \max_{\alpha+\rho \leq t \leq \beta-\rho} |\widetilde{F}_{t,h}^{(\alpha,\beta]}(u_m)|, D_m^{\mathcal{I}} \leftarrow \arg\max_{\alpha+\rho \leq t \leq \beta-\rho} |\widetilde{F}_{t,h}^{(\alpha,\beta]}(u_m)|$

    **else**

      $(A_m^{\mathcal{I}}, D_m^{\mathcal{I}}) \leftarrow (-1, 0)$

    **end if**

  **end for**

  $(m^*, \mathcal{I}^*) \leftarrow \arg\max_{m=1, \ldots, \lceil \log(T) \rceil, \mathcal{I} \in \mathcal{J}} A_m^{\mathcal{I}}$.

  **if** $A_{m^*}^{\mathcal{I}^*} > \tau$ **then**

    $\mathbf{S} \leftarrow \mathbf{S} \cup D_{m^*}^{\mathcal{I}^*}$

    FSBS $((s, D_{m^*}^{\mathcal{I}^*}], \bar{h}, h, \tau)$

    FSBS $((D_{m^*}^{\mathcal{I}^*}, e], \bar{h}, h, \tau)$

  **end if**

**OUTPUT:** The set of estimated change-points $\mathbf{S}$.

---

Algorithm 1 is conducted in an iterative way, starting with the whole time course, using the multi-scale seeded intervals to search for the point according to the largest CUSUM value. A change-point is declared if the corresponding maximum CUSUM value exceeds a pre-specified threshold $\tau$ and the whole sequence is then split into two with the procedure being carried on in the sub-intervals.

Algorithm 1 utilizes a collection of random grid points $\{u_m\}_{m=1}^{\lceil \log(T) \rceil} \subseteq \{x_{t,i}\}_{t=1, i=1}^{T, n}$ to detect changes in the functional data. For a change of mean functions at the time point $\eta$ with $\|f_{\eta+1}^* - f_\eta^*\|_\infty > 0$, we show in the appendix that, as long as $\lceil \log(T) \rceil$ grid points are sampled, with high probability, there is at least one point $u_{m'} \in \{u_m\}_{m=1}^{\lceil \log(T) \rceil}$ such that $|f_{\eta+1}^*(u_{m'}) - f_\eta^*(u_{m'})| \asymp \|f_{\eta+1}^* - f_\eta^*\|_\infty$. Thus, this procedure allows FSBS to detect changes in the mean functions without evaluating functions on a dense lattice grid and thus improves computational efficiency.

# 3 Main Results

## 3.1 Assumptions and theory

We begin by imposing assumptions on the kernel function $K(\cdot)$ used in FSBS.

**Assumption 2** (Kernel function). *Let $K(\cdot) : \mathbb{R}^d \to \mathbb{R}^+$ be compactly supported and satisfy the following conditions.*

**a.** *The kernel function $K(\cdot)$ is adaptive to the Hölder class $\mathcal{H}^r(L)$, i.e. for any $f \in \mathcal{H}^r(L)$, it holds that $\sup_{x \in [0,1]^d} \left| \int_{[0,1]^d} K_h(x-z) f(z) \, \mathrm{d}z - f(x) \right| \leq \tilde{C} h^r$, where $\tilde{C} > 0$ is a constant that only depends on $L$.*

**b.** *The class of functions $\mathcal{F}_K = \{K(x - \cdot)/h : \mathbb{R}^d \to \mathbb{R}^+, h > 0\}$ is separable in $\mathcal{L}_\infty(\mathbb{R}^d)$ and is a uniformly bounded VC-class. This means that there exist constants $A, \nu > 0$ such that for every probability measure $Q$ on $\mathbb{R}^d$ and every $u \in (0, \|K\|_\infty)$, it holds that $\mathcal{N}(\mathcal{F}_K, \mathcal{L}_2(Q), u) \leq (A\|K\|_\infty/u)^\nu$, where $\mathcal{N}(\mathcal{F}_K, \mathcal{L}_2(Q), u)$ denotes the $u$-covering number of the metric space $(\mathcal{F}_K, \mathcal{L}_2(Q))$.*

Assumption 2 is a standard assumption in the nonparametric literature, see [15, 16], [18], [27] among many others. These assumptions hold for most commonly used kernels, including uniform, polynomial and Gaussian kernels.

Recall the minimal spacing $\Delta = \min_{k=1}^{K+1} (\eta_k - \eta_{k-1})$ defined in Assumption 1**b**. We further define the jump size at the $k$th change-point as $\kappa_k = \|f^*_{\eta_k+1} - f^*_{\eta_k}\|_\infty$ and define $\kappa = \min_{k=1}^K$ as the minimal jump size. Assumption 3 below details how strong the signal needs to be in terms of $\kappa$ and $\Delta$, given the grid size $n$, the number of functional curves $T$, smoothness parameter $r$, dimensionality $d$ and moment condition $q$.

**Assumption 3** (Signal-to-noise ratio, SNR). *There exists an arbitrarily-slow diverging sequence $C_{\mathrm{SNR}} = C_{\mathrm{SNR}}(T)$ such that*

$$\kappa\sqrt{\Delta} > C_{\mathrm{SNR}} \log^{\max\{1/2, 5/q\}}(T)\left(1 + T^{\frac{d}{2r+d}} n^{\frac{-2r}{2r+d}}\right)^{1/2}.$$

We are now ready to present the main theorem, showing the consistency of FSBS.

**Theorem 1.** *Under Assumptions 1, 2 and 3, let $\{\widehat{\eta}_k\}_{k=1}^{\widehat{K}}$ be the estimated change-points by FSBS detailed in Algorithm 1 with data $\{x_{t,i}, y_{t,i}\}_{t=1, i=1}^{T, n}$, bandwidth parameters $\bar{h} = C_{\bar{h}}(Tn)^{-\frac{1}{2r+d}}$, $h = C_h(Tn)^{\frac{-1}{2r+d}}$ and threshold parameter $\tau = C_\tau \log^{\max\{1/2, 5/q\}}(T)\left(1 + T^{\frac{d}{2r+d}} n^{\frac{-2r}{2r+d}}\right)^{1/2}$, for some absolute constants $C_{\bar{h}}, C_h, C_\tau > 0$. It holds that*

$$\mathbb{P}\left\{\widehat{K} = K; |\widehat{\eta}_k - \eta_k| \leq C_{\mathrm{FSBS}} \log^{\max\{1, 10/q\}}(T)\left(1 + T^{\frac{d}{2r+d}} n^{\frac{-2r}{2r+d}}\right) \kappa_k^{-2}, \forall k = 1, \dots, K\right\}$$
$$\geq 1 - 3\log^{-1}(T),$$

*where $C_{\mathrm{FSBS}} > 0$ is an absolute constant.*

In view of Assumption 3 and Theorem 1, we see that with properly chosen tuning parameters and with probability tending to one as the sample size $T$ grows, the output of FSBS estimates the correct number of change-points and

$$\max_{k=1}^K |\widehat{\eta}_k - \eta_k|/\Delta \lesssim \left(1 + T^{\frac{d}{2r+d}} n^{\frac{-2r}{2r+d}}\right) \log^{\max\{1, 10/q\}}(T)/(\kappa^2 \Delta) = o(1),$$

where the last inequality follows from Assumption 3. The above inequality shows that there exists a one-to-one mapping from $\{\widehat{\eta}_k\}_{k=1}^K$ to $\{\eta_k\}_{k=1}^K$, assigning by the smallest distance.

## 3.2 Discussions on functional seeded binary segmentation (FSBS)

**From sparse to dense regimes.** In our setup, each curve is only observed at $n$ discrete points and we allow the full range of choices of $n$, representing from sparse to dense scenarios, all accompanied with consistency results. In the most sparse case $n = 1$, Assumption 3 reads as $\kappa\sqrt{\Delta} \gtrsim T^{d/(4r+2d)} \times$ a logarithmic factor, under which the localisation error is upper bounded by $T^{d/(2r+d)}\kappa^{-2}$, up to

a logarithmic factor. To the best of our knowledge, this challenging case has not been dealt in the existing change-point detection literature for functional data. In the most dense case, we can heuristically let $n = \infty$ and for simplicity let $q = \infty$ representing the sub-Gaussian noise case. Assumption 3 reads as $\kappa\sqrt{\Delta} \asymp \log^{1/2}(T)$ and the localisation error is upper bounded by $\kappa^{-2}\log(T)$. Both the SNR ratio and localisation error are the optimal rate in the univariate mean change-point localisation problem [28], implying the optimality of FSBS in the dense situation.

**Tuning parameters.** There are three tuning parameters involved. In the CUSUM statistic (3), we specify that the density estimator of the sampling distribution is a kernel estimator with bandwidth $\bar{h} \asymp (Tn)^{-1/(2r+d)}$. Due to the independence of the observation grids, such a choice of the bandwidth follows from the classical nonparametric literature [e.g. 26] and is minimax-rate optimal in terms of the estimation error. For completeness, we include the study of $\hat{p}(\cdot)$'s theoretical properties in **??**. In practice, there exist different default methods for the selection of $\bar{h}$, see for example the function `Hpi` from the R package `ks` ([9]).

The other bandwidth tuning parameter $h$ is also required to be $h \asymp (Tn)^{-1/(2r+d)}$. Despite that we allow for physical dependence in both functional noise and measurement error, we show that the same order of bandwidth (as $\bar{h}$) is required under Assumption 1. This is an interesting finding, if not surprising. This particular choice of $h$ is due to the fact that the physical dependence put forward by [30] is a short range dependence condition and does not change the rate of the sample size.

The threshold tuning parameter $\tau$ is set to be a high-probability upper bound on the CUSUM statistics when there is no change-point and is in fact of the form $\tau = C_\tau \log^{\max\{1/2,5/q\}}(T)\sqrt{n^{-1}h^{-d}+1}$. This also reflects the requirement on the SNR detailed in Assumption 3, that $\kappa\sqrt{\Delta} \gtrsim \tau$.

**Phase transition.** Recall that the number of curves is $T$ and the number of observations on each curve is $n$. The asymptotic regime we discuss is to let $T$ diverge, while allowing all other parameters, including $n$, to be functions of $T$. In Theorem 1, we allow a full range of cases in terms of the relationship between $n$ and $T$. As a concrete example, when the smooth parameter $r = 2$, the jump size $\kappa \asymp 1$ and in the one-dimensional case $d = 1$, with high probability (ignoring logarithmic factors for simplicity),

$$\max_{k=1}^{K} |\widehat{\eta}_k - \eta_k| = O_p(T^{\frac{1}{5}}n^{-\frac{4}{5}} + 1) = \begin{cases} O_p(1), & n \geq T^{1/4}; \\ O_p(T^{\frac{1}{5}}n^{-\frac{4}{5}}), & n \leq T^{1/4}. \end{cases}$$

This relationship between $n$ and $T$ was previously demonstrated in the mean function estimation literature [e.g. 8, 32], where the observations are discretely sampled from independently and identically distributed functional data. It is shown that the minimax estimation error rate also possesses the same phase transition between $n$ and $T$, i.e. with the transition boundary $n \asymp T^{1/4}$, which agrees with our finding under the change-point setting.

**Physical dependence and heavy-tailedness** In Assumption 1**c** and **d**, we allow for physical dependence type temporal dependence and heavy-tailed additive noise. As we have discussed, since the physical dependence is in fact a short range dependence, all the rates involved are the same as those in the independence cases, up to logarithmic factors. Having said this, the technical details required in dealing with this short range dependence are fundamentally different from those in the independence cases. From the result, it might be more interesting to discuss the effect of the heavy-tail behaviours, which are characterised by the parameter $q$. It can be seen from the rates in Assumption 3 and Theorem 1 that the effect of $q$ disappears and it behaves the same as if the noise is sub-Gaussian when $q \geq 10$.

# 4 Numerical Experiments

## 4.1 Simulated data analysis

We compare the proposed FSBS with state-of-the-art methods for change-point detection in functional data across a wide range of simulation settings. Specifically, we compare with three competitors: BGHK in [6], HK in [17] and SN in [31]. All three methods estimate change-points via examining mean change in the leading functional principal components of the observed functional data. BGHK is designed for temporally independent data while HK and SN can handle temporal dependence via

the estimation of long-run variance and the use of self-normalization principle, respectively. All three methods require fully observed functional data. In practice, they convert discrete data to functional observations by using B-splines with 20 basis functions.

For the implementation of FSBS, we adopt the Gaussian kernel. Following the standard practice in kernel density estimation, the bandwidth $h$ is selected by the function $\mathtt{Hpi}$ in the R package $\mathtt{ks}$ ([9]). The tuning parameter $\tau$ and the bandwidth $h$ are chosen by cross-validation, with evenly-indexed data being the training set and oddly-indexed data being the validation set. For each pair of candidate $(h, \tau)$, we obtain change-point estimators $\{\widehat{\eta}_k\}_{k=1}^{\widehat{K}}$ on the training set and compute the validation loss $\sum_{k=1}^{\widehat{K}} \sum_{t \in [\widehat{\eta}_k, \widehat{\eta}_{k+1})} \sum_{i=1}^{n} \{(\widehat{\eta}_{k+1} - \widehat{\eta}_k)^{-1} \sum_{t=\widehat{\eta}_k+1}^{\widehat{\eta}_{k+1}} F_{t,h}(x_{t,i}) - y_{t,i}\}^2$. The pair $(h, \tau)$ is then chosen to be the one corresponding to the lowest validation loss.

We consider five different scenarios for the observations $\{x_{ti}, y_{ti}\}_{t=1,i=1}^{T,n}$. For all scenarios 1-5, we set $T = 200$. Given the dimensionality $d$, denote a generic grid point as $x = (x^{(1)}, \cdots, x^{(d)})$. Scenarios 1 to 4 are generated based on model (1). The basic setting is as follows.

- **Scenario 1 (S1)** Let $(n, d) = (1, 1)$, the unevenly-spaced change-points be $(\eta_1, \eta_2) = (30, 130)$ and the three distinct mean functions be $6\cos(\cdot)$, $6\sin(\cdot)$ and $6\cos(\cdot)$.

- **Scenario 2 (S2)** Let $(n, d) = (10, 1)$, the unevenly-spaced change-points be $(\eta_1, \eta_2) = (30, 130)$ and the three distinct mean functions be $2\cos(\cdot)$, $2\sin(\cdot)$ and $2\cos(\cdot)$.

- **Scenario 3 (S3)** Let $(n, d) = (50, 1)$, the unevenly-spaced change-points be $(\eta_1, \eta_2) = (30, 130)$ and the three distinct mean functions be $\cos(\cdot)$, $\sin(\cdot)$ and $\cos(\cdot)$.

- **Scenario 4 (S4)** Let $(n, d) = (10, 2)$, the unevenly-spaced change-points be $(\eta_1, \eta_2) = (100, 150)$ and the three distinct mean functions be $0$, $3x^{(1)}x^{(2)}$ and $0$.

For **S1-S4**, the functional noise is generated as $\xi_t(x) = 0.5\xi_{t-1}(x) + \sum_{i=1}^{50} i^{-1} b_{t,i} h_i(x)$, where $\{h_i(x) = \prod_{j=1}^{d} (1/\sqrt{2})\pi \sin(ix^{(j)})\}_{i=1}^{50}$ are basis functions and $\{b_{t,i}\}_{t=1,i=1}^{T,50}$ are i.i.d. standard normal random variables. The measurement error is generated as $\delta_t = 0.3\delta_{t-1} + \epsilon_t$, where $\{\epsilon_t\}_{t=1}^{T}$ are i.i.d. $\mathcal{N}(0, 0.5I_n)$. We observe the noisy functional data $\{y_{ti}\}_{t=1,i=1}^{T,n}$ at grid points $\{x_{ti}\}_{t=1,i=1}^{T,n}$ independently sampled from $\mathrm{Unif}([0,1]^d)$.

Scenario 5 is adopted from [31] for densely-sampled functional data without measurement error.

- **Scenario 5 (S5)** Let $(n, d) = (50, 1)$, the evenly-spaced change-points be $(\eta_1, \eta_2) = (68, 134)$ and the three distinct mean functions be $0$, $\sin(\cdot)$ and $2\sin(\cdot)$.

The grid points $\{x_{ti}\}_{i=1}^{50}$ are 50 evenly-spaced points in $[0, 1]$ for all $t = 1, \cdots, T$. The functional noise is generated as $\xi_t(\cdot) = \int_{[0,1]} \psi(\cdot, u)\xi_{t-1}(u)\,\mathrm{d}u + \epsilon_t(\cdot)$, where $\{\epsilon_t(\cdot)\}_{t=1}^{T}$ are independent standard Brownian motions and $\psi(v, u) = 1/3 \exp((v^2 + u^2)/2)$ is a bivariate Gaussian kernel.

**S1-S5** represent a wide range of simulation settings including the extreme sparse case **S1**, sparse case **S2**, the two-dimensional domain case **S4**, and the densely sampled cases **S3** and **S5**. Note that **S1** and **S4** can only be handled by FSBS as for **S1** it is impossible to estimate a function via B-spline based on one point and for **S4**, the domain is of dimension 2.

**Evaluation result**: For a given set of true change-points $\mathcal{C} = \{\eta_k\}_{k=1}^{K}$, we evaluate the accuracy of the estimator $\{\widehat{\eta}_k\}_{k=1}^{\widehat{K}}$ by the difference $|\widehat{K} - K|$ and the Hausdorff distance $d(\hat{\mathcal{C}}, \mathcal{C})$, defined by $d(\hat{\mathcal{C}}, \mathcal{C}) = \max\{\max_{x \in \hat{\mathcal{C}}} \min_{y \in \mathcal{C}} \{|x - y|\}, \max_{y \in \hat{\mathcal{C}}} \min_{x \in \mathcal{C}} \{|x - y|\}\}$. For $\hat{\mathcal{C}} = \varnothing$, we use the convention that $|\widehat{K} - K| = K$ and $d(\hat{\mathcal{C}}, \mathcal{C}) = T$.

For each scenario, we repeat the experiments 100 times and Figure 1 summarizes the performance of FSBS, BGHK, HK and SN. Tabulated results can be found in **??**. As can be seen, FSBS consistently outperforms the competing methods by a wide margin and demonstrates robust behaviour across the board for both sparsely and densely sampled functional data.

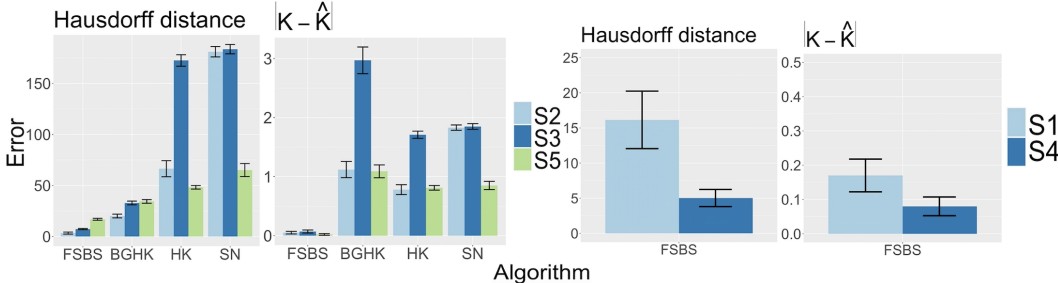

Figure 1: Bar plots for simulation results of **S1-S5**. Each bar reports the mean and standard deviation computed based on 100 experiments. From left to right, the first two plots correspond to the Hausdorff distance and $|K - \hat{K}|$ in **S2**, **S3** and **S5**. The last two plots correspond to **S1** and **S4**.

## 4.2 Real data application

We consider the COBE-SSTE dataset [24], which consists of monthly average sea surface temperature (SST) from 1940 to 2019, on a 1 degree latitude by 1 degree longitude grid ($48 \times 30$) covering Australia. The specific coordinates are latitude 10S-39S and longitude 110E-157E.

We apply FSBS to detect potential change-points in the two-dimensional SST. The implementation of FSBS is the same as the one described in Section 4.1. To avoid seasonality, we apply FSBS to the SST for the month of June from 1940 to 2019. We further conduct the same analysis separately for the month of July for robustness check.

For both the June and July data, two change-points are identified by FSBS, Year 1981 and 1996, suggesting the robustness of the finding. The two change-points might be associated with years when both the Indian Ocean Dipole and Oceanic Niño Index had extreme events [2]. The El Niño/Southern Oscillation has been recognized as an important manifestation of the tropical ocean-atmosphere-land coupled system. It is an irregular periodic variation in winds and sea surface temperatures over the tropical eastern Pacific Ocean. Much of the variability in the climate of Australia is connected with this phenomenon [5].

To visualize the estimated change, Figure 2 depicts the average SST before the first change-point Year 1981, between the two change-points, and after the second change-point Year 1996. The two rows correspond to the June and July data, respectively. As we can see, the top left corners exhibit different patterns in the three periods, suggesting the existence of change-points.

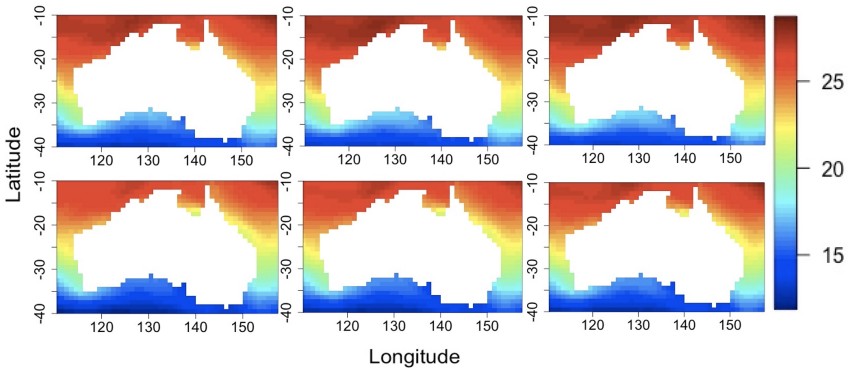

Figure 2: Average SST. From left to right: average SST from 1940 to 1981, average SST from 1982 to 1996, and average SST from 1997 to 2019. The top and bottom rows correspond to the June and July data respectively.

# 5   Conclusion

In this paper, we study change-point detection for sparse and dense functional data in general dimensions. We show that our algorithm FSBS can consistently estimate the change-points even in the extreme sparse setting with $n = 1$. Our theoretical analysis reveals an interesting phase transition between $n$ and $T$, which has not been discovered in the existing literature for functional change-point detection. The consistency of FSBS relies on the assumption that the minimal spacing $\Delta \asymp T$. To relax this assumption, we may consider increasing $\mathcal{K}$ in Definition 1 to enlarge the coverage of the seeded intervals in FSBS and apply the narrowest over threshold selection method [Theorem 3 in 19]. With minor modifications of the current theoretical analysis, the consistency of FSBS can be established for the case of $\Delta \ll T$. Since such a relaxation does not add much more methodological insights to our paper, we omit this additional technical discussion for conciseness.

## Acknowledgments and Disclosure of Funding

YY's research is partially funded by EPSRC EP/V013432/1.

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
