# OpenReview forum: "Change-point Detection for  Sparse and Dense Functional Data in General Dimensions"
_NeurIPS.cc/2022/Conference — NeurIPS 2022 Accept_

### Official Review · Reviewer_Kngx · 2022-07-10

**Rating:** 6
**Confidence:** 1
**Soundness:** 3 good
**Presentation:** 3 good
**Contribution:** 3 good

**Summary:**

This paper presented a change-point detection method, namely functional seeded binary segmentation (FSBS), for d-dimensional time-series data. FSBS is claimed to be more suitable for heavy-tailed and temporally dependent data, while being computationally efficient. The proposed method is evaluated on both synthetic data and sea surface temperature data and leads to better results compared to three classic methods: BGHK, HK, and SN.

**Questions:**

FSBS is claimed to handle temporally-dependent observations better theoretically, however, it is not clearly why it has such a property and other methods do not. In line 128 – 130, authors state that “For temporal dependence, we adopt the physical dependence framework by [26], which covers a wide range of time series models, such as ARMA and vector AR models.” while it is hard to tell how [26] is adopted here exactly.

**Ethics Review Area:**

["I don’t know"]

**Limitations:**

See “weakness” above.

**Strengths And Weaknesses:**

* Strengths

-- The proposed method is easy to implement with fewer (only three) tuning parameters, and the authors explained them well in terms of how to the select them.

-- The authors showed that FSBS is consistent in detecting and localising multiple change-points under standard regularity conditions. Besides, it is claimed to have the sharpest localisation error rate.

* Weaknesses

-- The baseline methods appear to be published quit a while ago, e.g., BGHK (2009), HK (2010) and SN (2011), so I’m not sure if they are state-of-the-art methods for change-point detection in functional data as the authors claimed in Line 267.

-- The experiments seem to be quite lightweight, as it has only one real dataset.

---

> ### Author Response · Authors · 2022-08-01
> **Responses**
>
> Thank you very much for your comments and suggestions. In the following, we reply to your comments point-by-point. We have submitted revised main text and supplementary files.
>
> **Whether competing method is state-of-the-art**
>
> To the best of our knowledge, the three competing methods are still the state-of-the-art in the literature of change-point estimation for functional data.
>
> **More real data analysis**
>
> In the revision, we have added one more real data example, considering with the same COBE-SSTE dataset as the real data set we have in our submission, using the same months June and July.  FSBS is applied to detect potential change points on a 1 degree latitude by 1 degree longitude grid $(10 \times 6)$, located at the Caribbean sea. In both months, FSBS identified the year $2004$ as a change-point. This might be associated with the development of a Modoki El Niño -– a rare type of El Niño in which unfavorable conditions are produced over the eastern Pacific instead of the Atlantic basin due to warmer sea surface temperatures farther west along the equatorial Pacific. Variability in the climate of northeastern Caribbean is connected with this phenomenon.  We have added plots in our revised supplementary materials.
>
> **Choice of method and the temporal dependence assumption**
>
> Our proposed FSBS algorithm is shown to be able to handle temporally-dependent observations, owing to our theoretical techniques.  We believe for other variants of the binary segmentation, our proposed theoretical techniques can be extended to show their respective theoretical performances when handling temporally-dependent data.  This is in fact a byproduct our paper - we provided peer researchers with new theoretical tools.  As for our paper, we chose the seeded binary segmentation method due to its computational efficiency.
>
> As for the physical dependence framework we adopted, it is used to model the dependence in the functional and scalar noise sequences, detailed in Assumption 1 (c) and (d).  The presence of the physical dependence inflates the signal-to-noise ratio condition, through the quantity $q$ in Assumption 3, and the localization error rate, again through the quantity $q$ in Theorem 1.

---

### Official Review · Reviewer_eNoD · 2022-07-11

**Rating:** 7
**Confidence:** 3
**Soundness:** 3 good
**Presentation:** 3 good
**Contribution:** 3 good

**Summary:**

This paper studies offline change-point detection and estimation for functional data. A kernel-based algorithm named functional seeded binary segmentation (FSBS) is proposed. This paper also shows the consistency of FSBS for multiple change-point estimation and 	.	provides a sharp localization error rate. Numerical demonstrations on synthetic and real data are provided.

**Questions:**

Some notations are not explained sufficiently. For example, what is the meaning of “arbitrarily-slow” in assumption 3, and what does C_SNR mean here?

Also, considering kernel-based change-point detection, it might be worthwhile to compare to or comment on the difference with some existing kernel type methods, such as "Li, S., Xie, Y., Dai, H., & Song, L. (2015). M-statistic for kernel change-point detection. Advances in Neural Information Processing Systems, 28." (although this paper didn't target detection for functional data, its online detection scheme might be helpful if the FSBS could be generalized for online detection).

Minor issues:
The equation in Theorem 1 is too long, maybe split into two lines.


**Limitations:**

yes

**Strengths And Weaknesses:**

Strengths

1. A new multiple change point detection algorithm is proposed. The FSBS procedure, based on kernel density estimation and CUSUM statistics, is computationally efficient.
2. This paper considers a general detection scheme, where the functional data can be discretely observed (even with n=1 in the very extreme case) and can be contaminated with temporally-dependent and heavy-tailed errors.
3. Theoretical guarantee on the estimation error is provided.
4. Extensive numerical experiments


Weaknesses
The method is not new. Both Binary Segmentation and kernel density estimations are well-studied methods for change detection and nonparametric density estimation, respectively. And the theoretical guarantee on BS and KDE are also well-studied.

---

> ### Author Response · Authors · 2022-08-01
> **Responses**
>
> Thank you very much for your comments and suggestions. In the following, we reply to your comments point-by-point. We have submitted revised main text and supplementary files.
>
> **Novelty**
>
> You are indeed right that our algorithm is based on existing methods. We are in fact proud of using existing and simple methods to achieve superior theoretical performance.  We would like to point out that although theoretical guarantees on binary segmentation and kernel density estimation are well-studied, it is the first time seen that such a combination (especially a newly-proposed variant of binary segmentation, namely seeded binary segmentation) is adopted for functional data analysis framework. Though our algorithm is simple, it can handle change-point estimation in discretely observed functional data (for both sparse and dense setting) with temporal dependence and heavy-tailed measurement errors. To our best knowledge, this is the first algorithm that can achieve such tasks in the literature.
>
> **Notation**
>
> By arbitrarily slow, we mean that the sequence $C_{\mathrm{SNR}}(T)$ diverges to infinity as $T$ grows unbounded and the divergence can be of any rate.  In other words, Assumption 3 that
> \begin{equation}
>     \kappa \sqrt{\Delta} > C_{\mathrm{SNR}} \log^{ \max\{1/2, 5/q\}}(T)\Big(1 + T^{\frac{d}{2r+d}} n^{\frac{-2r}{2r+d}} \Big)^{1/2}
> \end{equation}
> can be understood as
> \begin{equation}
>     \frac{\kappa \sqrt{\Delta}}{\log^{ \max\{1/2, 5/q\}}(T)\Big(1 + T^{\frac{d}{2r+d}} n^{\frac{-2r}{2r+d}} \Big)^{1/2}} \to \infty, \quad T \to \infty.
> \end{equation}
> We adopt the form in the first equation display instead of the second for the sake of finite sample results we present in Theorem 1.
>
> **More literature review**
>
> Thank you very much for your suggestion. In the literature, kernel based change-point detection is mainly used in nonparametric change-point estimation. Specifically, the goal is to locate change-points in the distribution of a sequence of data. See representative works such as Li et al. (2015), Arlot et al. (2019) and Li et al. (2019), Padilla et al. (2021).
>
> We remark that the goal of the aforementioned papers is very different from ours. The aforementioned papers all focused on nonparametric online or offline change point detection for independent time series data, while our method is designed to estimate change points for functional time series data with the presence of temporal dependence. In the rebuttal revision, we have added more discussions on this matter following your suggestion.
>
> **Reference**
>
> Sylvain Arlot, Alain Celisse, and Zaid Harchaoui. A kernel multiple change-point algorithm via model selection. Journal of Machine Learning Research, 20(162), 2019.
>
> Shuang Li, Yao Xie, Hanjun Dai, and Le Song. M-statistic for kernel change-point detection. Advances in Neural Information Processing Systems, 28, 2015.
>
> Shuang Li, Yao Xie, Hanjun Dai, and Le Song. Scan b-statistic for kernel change-point detection. Sequential Analysis, 38(4):503–544, 2019.
>
> Oscar Hernan Madrid Padilla, Yi Yu, Daren Wang, and Alessandro Rinaldo. Optimal nonparametric multivariate change point detection and localization. IEEE Transactions on Information Theory, 2021.

---

### Official Review · Reviewer_WXXv · 2022-07-11

**Rating:** 7
**Confidence:** 3
**Soundness:** 4 excellent
**Presentation:** 3 good
**Contribution:** 4 excellent

**Summary:**

This paper considers the change-point detection problem for functional data in multi-dimensional space. A kernel-based algorithm is proposed, which allows for handling functional data in multivariate domain with measurement error. The author further presents several important theoretical results about this algorithm, such as the consistency of testing, and sharp localization error rate. Numerical results support their contributions.

**Questions:**

See **Weakness** part.

**Limitations:**

See **Weakness** part.

**Strengths And Weaknesses:**

**Originality/Significance**: The proposed kernel-based algorithm elegantly solves main challenges in existing literature: (i) observed funtional data may contain measurement error, (ii) there may exist multiple change points, (iii) domain for functional data can be multivariate.  This work would be very interesting for readers in the testing and detection area.

**Quality**: All important points for functional-data change point detection are discussed, including the algorithm development, solid theoretical analysis, and extensive experiment simulations. From my view point, the quality of this paper is at least above 50% of all accepted nips papers.

**Clarity**: Problem setup, algorithm procedures, theory statements and assumptions are clearly presented.

**Weakness**:
1. It would be helpful to do a literature review for kernel-based change-point detection (even not under the functional data setting). This comparison will make the reader understand the connection and difference for the proposed algorithm.
2. The choice of kernel will largely influence the performance of change-point detection. The kernel choice in general is specified by the users that have prior knowledge on the data. Although the author does not demonstrate which kernel choice is better (in numerical simulation only Gaussian kernel is used), I think it is not a big issue and it would be an interesting topic for future study.
3. Minor wording issue:
- In line 168, the bracket should be $O(T\log(T))$.
- In line 211-212, there is a large box for the equation environment. The author may consider repharse it to be "with probability at least $1-3\log^{-1}(T)$, it holds that \[xxx\]".

---

> ### Author Response · Authors · 2022-08-01
> **Responses**
>
> Thank you very much for your comments and suggestions. In the following, we reply to your comments point-by-point. We have submitted revised main text and supplementary files.
>
> **More literature review**
>
> Thank you very much for your suggestion.  In the literature, kernel based change-point detection is mainly used in nonparametric change-point estimation. Specifically, the goal is to locate change-points in the distribution of a sequence of data.  See representative works such as Li et al. (2015), Arlot et al. (2019) and Li et al. (2019), Padilla et al. (2021).
>
> We remark that the goal of the  aforementioned papers  is very different from ours. The aforementioned papers all focused on nonparametric online or offline change point detection for independent time series data, while our method is designed to estimate change points for functional time series data with the presence of temporal dependence.   In the rebuttal revision, we have added more discussions on this matter following your suggestion.
>
> **Choice of kernels**
>
> You are right that the choice of kernels may affect the performance of any kernel based method. We choose Gaussian kernel for demonstration as it is one of the most popular kernel used in practice. Following your suggestion, we also tried other kernels in our simulation study and we find that the numerical performance of our method is robust to the choice of kernels.  Tables below summarize results of the performance of the FSBS with different choices of kernels on Scenarios 1 and 2 in the paper. Here we consider common kernels used in literature as Gaussian, Uniform, Epanechnikov and Quartic.  We have added the additional simulation results in the revised supplementary materials.
>
> Scenario 1
> |Kernel|$K- \hat{K}<0$|$K- \hat{K}=0$|$K- \hat{K}>0$|$\vert \hat{K}-K\vert$|Haus. dist|
> |--|--|--|--|--|--|
> |Gaussian|0.05|0.86|0.09|0.17 (0.05)|16.15 (4.09)|
> |Uniform|0.01|0.99|0|0.01 (0.01)|13.32 (0.42)
> |Epanechnikov|0.06|0.87|0.17|0.13 (0.03)|15.14 (2.40)|
> |Quartic|0.07|0.84|0.09|0.20 (0.04)|18.28 (1.63)||
>
> Scenario 2
> |Kernel|$K- \hat{K}<0$|$K- \hat{K}=0$|$K- \hat{K}>0$|$\vert \hat{K}-K\vert$|Haus. dist|
> |--|--|--|--|--|--|
> |Gaussian|0.05|0.95|0|0.05 (0.02)|3.32 (1.00)|
> |Uniform|0|0.99|0.01|0.01 (0.01)|2.93 (1.03)|
> |Epanechnikov|0|1|0|0 (0)|1.24 (0.28)|
> |Quartic|0.01|0.99|0|0.01 (0.01)|2.30 (0.55)|
>
>
> The mean is reported in these tables. We obtain the results over 100 repetitions and the numbers in parenthesis denote standard errors.
>
> **Typos**
>
> Thank you very much for pointing these out.  In the revision, we have corrected all these typos.
>
> **Reference**
>
> Sylvain Arlot, Alain Celisse, and Zaid Harchaoui. A kernel multiple change-point algorithm via model selection. Journal of Machine Learning Research, 20(162), 2019.
>
> Shuang Li, Yao Xie, Hanjun Dai, and Le Song. M-statistic for kernel change-point detection. Advances in Neural Information Processing Systems, 28, 2015.
>
> Shuang Li, Yao Xie, Hanjun Dai, and Le Song. Scan b-statistic for kernel change-point detection. Sequential Analysis, 38(4):503–544, 2019.
>
> Oscar Hernan Madrid Padilla, Yi Yu, Daren Wang, and Alessandro Rinaldo. Optimal nonparametric multivariate change point detection and localization. IEEE Transactions on Information Theory, 2021.

---

### Official Review · Reviewer_3KgB · 2022-07-12

**Rating:** 8
**Confidence:** 3
**Soundness:** 3 good
**Presentation:** 4 excellent
**Contribution:** 4 excellent

**Summary:**

This paper proposes a new kernal-based algorithm called functional seeded binary segmentation (FSBS) for change point detection of functional data. Theoretical error rate and consistency results are presented for functional data with general sampling density (sparse & dense), temporal dependence, and importantly that are multi-dimensional. Numerial experiments are conducted on simulation data as well as climate change data.

**Questions:**

1. Since the paper stresses extension to multi-dimensional data as a major contribution, there should be more discussion (in section 3.2) and experiments (in both section 4.1 and 4.2) involving multi-dimensional data. What insight can one gain about the theoretical bounds with respect to parameter $d$? How does the algorithm behave for $d >1$ in more than just a single experiment (scenario 4)?

2. How did the computational cost and runtime of the proposed method compare to the existing methods?

3. It's not clear what the reader is supposed to notice from Figure 2. If there is noticable change in the top left corner, a zoomed-in panel might be helpful.


**Strengths And Weaknesses:**

This paper is very clearly written and the presentation is great. While I haven't had a chance to read through the proofs in appendix, assuming they are technically sound, extending changepoint detection to multi-dimensional functional data is a significant contribution to the community.

---

> ### Author Response · Authors · 2022-08-01
> **Responses**
>
> Thank you very much for your comments and suggestions. In the following, we reply to your comments point-by-point. We have submitted revised main text and supplementary files.
>
> **Theoretical comments on the dimension $d$**
>
> We first discuss dimension $d$ in terms of the theory. Recall that the localization error rate of change-point estimation in Theorem 1 is $C_{\mathrm{FSBS}} \log^{ \max\{1, 10/q \} }  (T)  \left(1 +   T^{\frac{d}{2r+d}}   n^{\frac{-2r}{2r+d}}  \right) \kappa_k^{-2}.$ Holding everything else the same, it can be clearly seen that $T^{\frac{d}{2r+d}}   n^{\frac{-2r}{2r+d}}$ is an increasing function of $d$, i.e. a larger $d$ will lead to a worse localization error rate.
>
> In addition, recall that in Assumption 3 we require the signal-to-noise ratio to be lower bounded by $C_{\mathrm{SNR}} \log^{ \max\{1/2, 5/q\}}(T)\Big(1 + T^{\frac{d}{2r+d}} n^{\frac{-2r}{2r+d}} \Big)^{1/2}.$ Thus, a larger $d$ will also require a stronger SNR.
>
> **Additional numerical results on the dimension $d$**
>
> In the revision, we have analyzed the performance of FSBS with different choices of dimension $d$ for Scenario 4 in the manuscript, with  $d \in \{2, 3, 5, 10\}$ and all the other details identical to Scenario 4.  The results are collected in the table below with more additional details in the revised supplementary materials. These support the previous discussion on dimension $d$.
>
> | Dimension $d$|$K- \hat{K}<0$|$K- \hat{K}=0$|$K- \hat{K}>0$|$\vert\hat{K}-K\vert$|Haus. dist.|
> | ---- | --- |--- |--- |--- |--- |
> | 2 | 0|0.90|0.08|0.08 (0.02)|5.02 (1.25)|
> |3|0.02|0.89|0.09|0.11 (0.03)|5.73 (1.22)|
> |5|0.18|0.82|0|0.18 (0.05)|5.92 (1.23)|
> |10|0.21|0.79|0|0.22 (0.08)|6.58 (1.24)|
>
> The mean is reported in this table. We obtain the results over 100 repetitions and the numbers in parenthesis denote standard errors.
>
> **Computation time**
>
> Our method is computationally efficient and its computational complexity is $O(nT\log T+T(\log T)^2)$. Specifically, as can be seen from Algorithm 1, we need to conduct kernel smoothing of the sampling distribution and mean function at $\log T$ measurement locations, which costs $O(nT\log T)$ operations. Once this is done, we conduct seeded binary segmentation (SBS) at the $\log T$ measurement locations/grids. It is known that SBS has a computational cost of $O(T\log T)$. Thus, this step costs $O(T(\log T)^2)$ computational complexity. In total, the computational complexity of our method is $O(nT\log T+T(\log T)^2)$.
>
> As for existing methods in the literature, in terms of implementation, they all rely on the two-stage procedure. Specifically, the first stage is to register/estimate the discretely observed points into a functional curve at each time $t$. Taking the B-spline smoothing with $p$ basis functions for example, this costs $O(n^2p+p^3)$ computational complexity for each time $t$ due to a least square estimation. Thus this step costs $O(T(n^2p+p^3))$ computational complexity. Once the functional curves are registered, in the second stage, the existing methods conduct functional PCA to extract $p'$ principle component scores from each function and then conduct mean change-point detection on the $p'$-dimension time series of principle component scores. Ignoring the computational cost of functional PCA, the change-point detection procedure costs at least $O(T\log T)$ computational complexity if a standard binary segmentation is used and could be more expensive if other segmentation algorithms are used to conduct change-point estimation. Thus, in total, the computational complexity of existing methods is at least $O(T(n^2p+p^3)+T\log T)$, which is more expensive unless $n\preceq\log T.$
>
> We have added this discussion in Section A.1 in the supplementary materials in the rebuttal revision, following your suggestions.
>
> **Figure 2**
>
> Thanks for your suggestion.  In the revised supplementary materials, we have added a zoomed-in version of Figure 2, emphasizing the northwest and northeast coasts of Australia respectively.

---

### Official Review · Reviewer_1qwN · 2022-07-18

**Rating:** 4
**Confidence:** 4
**Soundness:** 2 fair
**Presentation:** 3 good
**Contribution:** 2 fair

**Summary:**

This paper studies change-point detection and localization for sparsely or densely sampled functional data observed on a general d-dimensional space. A computational efficient kernel based FSBS algorithm is proposed to handle discretely observed functional data and general d-dimensional domain. The supporting theory is developed for heavy-tailed and serially dependent curves and measurement errors.  A phase transition phenomenon is established depending on the order of n relative to T. The finite sample performance is evaluated via both simulations under different settings and one real data example.

**Questions:**

1, What is the advantage of the local constant smoother used in the paper instead of the commonly adopted local linear smoother?

2, The phase transition result is interesting, but is different from results in Zhang and Wang, 2016, which considers categories of sparse, dense and ultra-dense functional data. In particular, the optimal bandwidths are different under different cases and as the number of measurement locations becomes sufficiently large, the parametric root-n rate can be attained. A detailed discussion is helpful.

3, What are the truly innovative component of the paper taking the advantages of functional, temporally dependent and non-stationary nature of the data?

4, Why is the extreme sparse case n=1 argued as one selling point of the paper? As far as I know, if n is bounded (treated as sparse functional data), the same rate can be attained.

**Ethics Review Area:**

["I don’t know"]

**Limitations:**

Yes

**Strengths And Weaknesses:**

Strength:

1, The first work in existing literature studying change points detection for partially observed functional data.

2, Can handle low-dimensional multivariate functional data.

3, Theory is developed for serially dependent random functions and measurement errors

Weakness:

1, Instead of a machine learning conference like NeurIPS, I feel this paper fits a traditional Statistics journal better. The proposed methodology seems like a combination of some well-established techniques in change points detection and functional data analysis areas.

2, The measurement locations x_{t,i} are fixed, while in standard functional and longitudinal data analysis literature, these locations are assumed to be random and the number of measurement locations across curves can be different (rather than a fixed n) or even random.

3, The detailed comparison with phase transition phenomenon in existing literature is needed. See my question below.

4, Local constant smoother instead of local linear smoother is used and hence may suffer from bad performance around the boundaries.

---

> ### Author Response · Authors · 2022-08-01
> **Responses**
>
> Thank you very much for your comments and suggestions.  In the following, we reply to your comments point-by-point.  We have submitted revised main text and supplementary files.
>
> **Observation grids**
>
> Measurement locations $\{x_{t,i}\}$ are usually assumed to be random. This is the case in our paper. In Assumption 1(a), we assume that the observation grids $\{x_{t, i}\}_{t = 1, i = 1}^{T, n}$ are independently sampled from a distribution, with certain regularity conditions.
>
> > (Discrete grids) The grids $x_{t,i} \in [0, 1]^d, t = 1, \ldots, T,  i = 1, \ldots, n,$ are independently sampled from a common  density function $u: [0,1]^d \to \mathbb R$. In addition, suppose   that there exist constants  $r>0$ and $L>0$ such that   $ u \in  \mathcal{H}^r (L)  $ and that  $\inf_{x\in [0,1]^d } u(x) \ge \tilde{c}$ with an absolute constant $\tilde{c}>0$.
>
> As mentioned in our manuscript, the assumption that different functional curves have the same number of grid points $n$ is made for presentation simplicity only.  Our theoretical results continue to hold under the following weaker assumption on the observation grids.
>
> > Different functional curves are assumed to have the same number of grid points $n$. We remark that this is made for presentation simplicity only. It can indeed be further relaxed and the main results below will then depend on both the minimum and maximum numbers of grid points.
>
> **Local constant smoother vs. local linear smoother**
>
> For estimation of a function, in general, local constant smoother indeed may not perform well around the boundaries. However, note that our primary focus here is *not* the estimation of a function, but estimating potential change-points. Our theory suggests that local constant smoother is sufficient for achieving optimality in the localization error rate of change-point estimation. Local constant smoother is computationally cheaper than local linear estimator, facilitating our fast computation. Once the change-points are localised by our method, we can use local linear smoother to further estimate the mean function within each estimated stationary segment.
>
> **Phase transition**
>
> Thank you for bring Zhang and Wang (2016) (ZW) to our attention, which concerns estimation of a function given a sequence of stationary functional data. The phase transition result in ZW matches the one in our paper. In the notation of our paper, ZW consider the special case of $r=2$ (smoothness of mean function) and $d=1$ (dimension of function domain).
>
> Compare their Corollary 3.2 and our Theorem 3.1. For $r=2$ and $d=1$, our bandwidth in Theorem 3.1 is $(Tn)^{-1/5}$, which exactly matches their bandwidth in case 1 (sparse) and case 2 (dense). For case 3 (ultra dense), ZW allows any bandwidth that is $o(T^{-1/4})$, which is satisfied by our bandwidth $(Tn)^{-1/5}$ under the ultra dense case ($n/T^{1/4}\to \infty$). This indeed further validates our theoretical results.
>
> As for the parametric rate (in our notation is parametric root-$T$ rate), our primary focus is change-point localization, and the well-known parametric rate for change-point localization (when dealing with a constant jump size $\kappa$) is $O_p(1)$, which is achieved by our method (up to logarithmic factors) under the case where $n\geq T^{1/4}$ (this is discussed in our manuscript).  The reason that we can achieve this parametric $O_p(1)$ rate for change-point localization is that under the case $n\geq T^{1/4}$, we can estimate the mean function up to the parametric root-$T$ rate.
>
> **Innovation**
>
> It is not that our method is taking advantage of temporally dependent and non-stationary nature of the functional data. Instead, our method and theory are designed to be capable of estimating multiple change-points for (discretely observed) functional data under temporal dependence and heavy-tailedness, which are more realistic assumptions than those used in the literature.
>
> Our proposed algorithm FSBS is the first methodology that can rigorously estimate the change-points even when functional data are sparsely observed. This desired feature has not been seen in the existing literature on functional change point detection. In the review by Wang, Chiou and Müller, sparse functional data are described as *much more difficult but commonly encountered situation* and *typically require more effort in theory and methodology as compared to densely sampled functional data*.
>
> **Sparse cases**
>
> You are right, a bounded constant $n$ and $n=1$ will give the same order of localization rate, which can be clearly seen from Theorem 1 in our paper. To our best knowledge, no existing works on change-point estimation in functional data can rigorously handle the case of a bounded constant $n$ (i.e. the sparse case), thus this is a selling point of our paper. We choose to advertise $n=1$ as it is the most representative case (and most challenging case in practice) of a bounded constant $n$.

---

### Meta-Review · Area_Chair_FuS4 · 2022-08-27

**Recommendation:** Accept
**Confidence:** Certain

**Metareview:**

The paper studies change-point detection and localization for functional data, which is an interesting and timely topic. I agree with some reviewers that the paper might be a better fit with the traditional statistical venue. The authors have done a great job in the rebuttal phase in addressing reviewers’ comments. I believe it is a worthwhile paper to be published in NeurIPS.

**Award:**

No

---

### Decision · Program_Chairs · 2022-09-14

Accept